# The Aroma Volatile in 'Nanguo' Pear: A Review

Zhuoran Zhang [1,2,3] and Zepeng Yin [1,2,3,*]

1   Key Laboratory of Fruit Postharvest Biology, Shenyang 110866, China
2   Key Laboratory of Protected Horticulture, National & Local Joint Engineering Research Center of Northern Horticultural Facilities Design & Application Technology, Shenyang 110866, China
3   College of Horticulture, Shenyang Agricultural University, Shenyang 110866, China
*   Correspondence: yinzp@syau.edu.cn

**Abstract:** The aroma of fruit is an important indicator that reflects the quality of its flavor. The 'Nanguo' pear (*Pyrus ussuriensis* Maxim.) is a typical fruit in Liaoning Province, China, that has an attractive aroma during fruit ripening. Fruit volatile compounds are primarily composed of esters, alcohols, aldehydes, ketones, lactones, terpenoids and apocarotenoids. The primary characteristic volatile compounds of the 'Nanguo' pear are esters. The contents of aldehydes decrease, and the contents of esters increase as the fruit ripens. The aroma changes from 'green' to 'fruity.' Thus, it has been a favorite of consumers in China and throughout the world for its attractive aroma. This review systematically summarizes advances in the research methods, components, types and biosynthetic pathways of volatile compounds, and the factors that affect the aroma volatiles in the 'Nanguo' pear, particularly the regulation by hormones that has been studied in recent years and delineates the research problems and prospects. The aim is to provide critical information for further research on the qualities of fruit flavor and provide a more scientific basis to improve the quality of fruit flavor during the development and storage of the 'Nanguo' pear.

**Keywords:** Aroma compounds; *Pyrus ussuriensis*; biosynthetic pathways; fruit quality

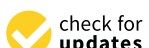



## 1. Introduction

Plants have the ability to synthesize, accumulate and release volatiles [1]. For plants, the reason for the release of volatile substances may be to spread seeds. For humans, these volatile compounds (VOCs) may have antibacterial or anticancer activity. People will also extract some of these ingredients to make perfume [2]. Fruit flavor plays an essential role in consumer preferences, including sensations in the mouth (sweetness, acidity, or bitterness) and odors produced by several VOCs [3,4]. Although sugar and acid are related to consumer preferences, it is the VOCs that determine the distinctive taste of ripe fruit [4,5].

The demand for high-quality fruit is increasing as living standards improve. Thus, aroma volatiles have become one of the criteria for consumer preferences. Fruits with attractive aromas will become more desirable for purchase by consumers [6,7]. However, the degradation of fruit quality, particularly the losses in aroma that often occur, reduce the demand of consumers to purchase them [3,8,9]. Therefore, studies on fruit aroma have become a hot topic, and researchers have begun to focus on this problem [10].

Considering the important influence of VOCs on flavor, a lot of research has been carried out on the VOCs of fruits. Breakthroughs have been made in the biosynthesis path of fruit aroma substances, the identification of key structural genes, and the regulation of exogenous hormones on the metabolism of aroma substances [1,11,12]. With the emergence of precision instruments such as gas chromatography-mass spectrometry (GC-MS), the research on aroma volatiles has become more in-depth, and the method for determining aroma volatiles has gradually matured.

'Nanguo' pear (*Pyrus ussuriensis* Maxim.) is a unique local pear species in Liaoning Province, China, that produces attractive aroma volatiles during fruit ripening and has

rarely been studied for its fruit aroma [13,14]. Freshly harvested 'Nanguo' pears are green with poor taste and mild aroma volatiles [15,16]. After moderate post-ripening, the fruit become yellow, rich in juice, and have an attractive aroma [14,17]. During the fruit ripening process of the 'Nanguo' pear, the content of aldehydes gradually decreases, and the content of esters gradually increases, which changes the aroma from 'green' to 'fruity' [18]. However, after post-ripening at room temperature, the quality of the 'Nanguo' pear will decrease sharply and lose its commercial value [13]. Cold storage is often used to prolong the shelf life of the 'Nanguo' pear in production, which results in reduced flavor, and a decrease in the appeal of the 'Nanguo' pear to consumers [13,16]. The 'Nanguo' pear is a good example to study fruit aroma. So far, 79 volatile chemicals have been observed in the 'Nanguo' pear. These compounds include esters, alcohols, ketones, terpenes, aldehydes and benzene compounds [8]. The types and concentrations of these volatiles vary according to their cultivation conditions and maturity [15,19]. The most important aroma volatiles include fatty acid-derived compounds and amino acid-derived compounds in the 'Nanguo' pear [8]. Although aroma volatiles are usually a complex mixture of a wide range of compounds, volatile esters often represent the major contribution in the 'Nanguo' pear.

In this review, extraction and identification methods, species and composition, the biosynthetic pathway, factors that influence aroma volatiles in the 'Nanguo' pear and their regulation by hormones were examined, which will provide a theoretical basis for the further development of this pear.

## 2. The Methods of Extracting and Identifying Volatiles from the 'Nanguo' Pear

The aromas of fruit are apparent to humans as olfactory substances, which are captured by the olfactory organs and passed to the olfactory nerve, thus, providing a pleasant feeling [6]. However, owing to limitations in the detection instrument, the understanding of aromas is restricted to sensory evaluation during the early phase. However, with the quick development of science and technology, particularly with the application of the GC-MS and other analytical apparatus, detection methods have improved, and some instruments can be used to enrich and analyze the components of aromas [20]. Currently, there are several methods to extract the aromatic volatiles of the 'Nanguo' pear. Each of them has their own characteristics and can be used selectively to obtain better extraction results.

### 2.1. Solvent Extraction (SE)

Most aromatic compounds can be extracted from materials by solvent extraction since they are highly soluble in some organic solvents [21]. SE requires a large amount of extractant when extracting VOCs that are present in low amounts [22]. SE often produces solvent pollutants, and the method is not very effective at extracting the VOCs from fruit [21,22]. Currently, it is no longer the primary method used to extract VOCs.

### 2.2. Static Headspace Extraction (SHS)

The method uses an extractant to release the samples, which are then heated to concentrate them before transfer to an equilibrium headspace bottle to extract some of the gases for analysis [23]. SHS is suitable for extracting aroma compounds that have a low boiling point and are highly volatile [23,24]. Zhang et al. [23] extracted 22 aroma components from the 'Nanguo' pear by the SHS method, including 14 esters, one alcohol, one aldehydes, two ketones, three phenolics and one hydrocarbon. Among them, ethyl caproate and α-farnesene were the most prevalent compounds.

### 2.3. Simultaneous Distillation Extraction (SDE)

An aqueous solution that contains the aroma volatiles and the extractant are simultaneously heated and boiled at atmospheric pressure, and the water vapor that contained the aroma volatiles is extracted to separate the compounds [16,25]. After heating and concentration, the solution is transferred to a chromatographic bottle to detect the components [23]. The method combines concentration and extraction to improve the efficiency of separation

and maintain the aroma components [25]. Zhang et al. [23] extracted 53 aroma components from 'Nanguo' pear using the SDE method, including 35 esters, five hydrocarbons, six alcohols, two aldehydes, two ketones, one acid and two phenolics. The relatively high contents included ethyl caproate, ethyl butyrate and α-farnesene.

*2.4. Headspace Solid Phase Microextraction (HS-SPME)*

This method integrates extraction, concentration and injection. A quartz fiber with the extraction layer is inserted into the sample or saturated air that contains the aroma volatiles, which are then analyzed by mass spectrometry [26]. This method is safe, simple, and has a high extraction rate, making it an ideal extraction method [21,27]. However, this method has a long detection period and expensive extraction fiber. It is more suitable for trace analysis; there will be a large error when determining a high content of aroma volatiles [21]. This is the most commonly used method to study the aroma components of the 'Nanguo' pear, which can effectively detect esters, aldehydes, and other substances in these fruits [8,28,29]. Yi [28] used this method to extract ten esters, two terpenoids and three hydrocarbons from the intact 'Nanguo' pear, including four characteristic aroma volatiles. These compounds were identified as ethyl caproate, ethyl butyrate, ethyl 2-methylbutyrate and hexyl caproate. We used this method to identify 79 VOCs in 'Nanguo' pears, including 4 aldehydes, 31 esters, 4 ketones, 1 alcohol, 50 terpenoids, 11 phenolic compounds, and 13 other substances. In addition, this method was used to identify three characteristic aroma volatiles of the 'Nanguo' pear, including ethyl hexanoate, hexyl acetate, and ethyl butanoate, which represented the dominant esters in the ripened pear fruit [8].

Using a single method to extract aroma substances often cannot meet the needs of the analysis. Currently, there is no good method to accurately reflect the actual aroma components or their proportions in fruits. The commonly used HS-SPME method cannot extract carboxylic acid compounds well, while the SDE can detect carboxylic acid compounds well, so SDE and HS-SPME can be used to analyze more complete aroma components [30]. When detecting the aroma components of the 'Nanguo' pear, SDE can obtain more complete aroma information, but because of its solvent pollution, the HS-SPME method is used more [23]. It is very important to find a faster and safer aroma determination method.

After the aroma components have been enriched, they can be detected by GC-MS and electronic nose technology [31–33]. Both methods have their advantages and disadvantages. GC-MS is a commonly used technique to identify aromatic components [8,16]. The results are more accurate and highly precise, but the process is more complex [31]. However, electronic nose technology can systematically and scientifically analyze and detect sample odors by simulating biological olfaction, while the species of detectable aroma compounds are limited owing to the small scale of the sensor array [33].

## 3. Category and Composition of the Primary Aroma Components in 'Nanguo' Pears

The aroma of fruit originates from VOCs, but not all VOCs can be perceived by humans [2]. The aromatic compounds will only contribute to the aroma of the fruit when the substance reaches a specific threshold [3]. Since the contribution of VOCs to aroma is affected by the aroma threshold, it is not easy to perceive its aroma when there is a large threshold [2]. The odor activity value (OAV) is usually used to measure the contribution of a substance to the aroma [34]. OAV is the ratio of the concentration of VOCs in food to their detection threshold. [35]. Only when the OAV value >1, are the VOCs regarded as significantly characteristic volatiles or key aroma compounds [28,35]. Higher OAV values of the VOCs indicate that they have a pivotal role.

More than 300 VOCs have been identified in pears [3]. A combination of SDE with GC-MS was used to study the VOCs of four Asian pears (*Pyrus serotina*) [36]. Eight esters and one aldehyde were extracted. Among them, ethyl 2-methylbutyrate is an important contributor to the aroma of four Asian pears and is usually found in pineapple (*Ananas comosus*) and green apples (*Malus domestica*) [36]. Our previous study identified nearly 100 VOCs using HS-SPME combined with GC-MS in 'Nanguo' pear fruit samples in both 2019

and 2021 [8]. These VOCs were divided into acids, hydrocarbons, esters, alcohols, ketones, and aldehydes [3,37].

HS-SPME combined with GC-MS was used to analyze the aroma volatiles of 'Nanguo' pears at different harvest times. Eight characteristic VOCs were identified in these pears, including five esters and three aldehydes, which were ethyl butyrate, butyl acetate, ethyl 2-methylbutyrate, ethyl hexanoate, hexyl acetate, hexanal, 2-hexenal and decanal. The content of aldehydes in early harvested 'Nanguo' pears increased, and their primary aroma components did not change, which may be caused by the short harvest interval [19]. Our study used HS-SPME combined with GC-MS and identified five types of VOCs in the 'Nanguo' pear, including aldehydes, alcohols, esters, terpenes, and phenolics. [8]. The typical fatty acid-derived VOCs differ in different ripening stages, which results in a shift in the composition of volatiles from aldehydes, alcohols, and ketones to esters and terpenes during pear ripening [13,18]. Esters are the primary aroma components in mature 'Nanguo' pear [38]. We detected the VOCs of 'Nanguo' pear within 15 days after harvest (DAH). Among them, ethyl butyrate, ethyl hexanoate and hexyl acetate are considered to be the most important typical aroma substances based on their high content and low odor threshold, which are used to characterize the aroma content of 'Nanguo' pear (Table 1) [28]. Previous studies have shown that there is a synergistic effect between the aroma of fruit. The fruit aroma is not only related to the content of certain characteristic VOCs but is also related to the species of characteristic VOCs [39]. However, the synergistic effect between aromas is not clear, and it is not possible to determine the relationship between the strong aroma and the synergistic effect of aroma in a 'Nanguo' pear. The aldehydes in pear fruit are primarily $C_6$ aldehydes, which have a grassy smell [40]. In 'Nanguo' pear, the content of esters is much higher than the content of aldehydes after ripening [38]. Currently, it is believed that the aroma of 'Nanguo' pears come from the typical aroma substances.

**Table 1.** Volatile compounds detected in 'Nanguo' Pears fruit during fruit ripening.

| No. | Compounds | 0 DAH ($\mu$g kg$^{-1}$) | 5 DAH ($\mu$g kg$^{-1}$) | 10 DAH ($\mu$g kg$^{-1}$) | 15 DAH ($\mu$g kg$^{-1}$) |
|---|---|---|---|---|---|
| | Esters | | | | |
| 1 | Ethyl Acetate | — | — | — | 180.82 ± 2.40 |
| 2 | Ethyl Butanoate | — | — | 352.58 ± 3.40 | 550.58 ± 5.40 |
| 3 | Butyl Acetate | — | — | 38.58 ± 0.96 | — |
| 4 | Ethyl Valerate | — | — | 51.10 ± 2.26 | 71.12 ± 3.92 |
| 5 | Methyl Hexoate | — | — | 81.62 ± 8.10 | 223.08 ± 16.28 |
| 6 | Ethyl Hexanoate | — | — | 1230.34 ± 49.12 | 3521.68 ± 472.54 |
| 7 | Hexyl Acetate | — | — | 845.88 ± 10.42 | 1082.00 ± 31.72 |
| 8 | Ethyl Hex-3-Enoate | — | — | 13.16 ± 0.52 | — |
| 9 | Heptyl Hexanoate | — | — | 24.72 ± 1.28 | — |
| 10 | Hex-2-Enoic Acid Ethyl Ester | — | — | — | 47.00 ± 0.74 |
| 11 | Heptylacetat | — | — | 14.98 ± 1.10 | 39.6 ± 1.22 |
| 12 | Methyl Octylate | — | — | — | 31.26 ± 1.18 |
| 13 | Hexyl Butyrate | — | 13.56 ± 0.26 | 20.44 ± 0.60 | — |
| 14 | Ethyl Caprylate | — | — | 51.54 ± 0.78 | 285.88 ± 8.42 |
| 15 | Ethyl (Z)-Oct-4-Enoate | — | — | 12.08 ± 0.68 | — |
| 16 | Octyl Acetate | — | — | — | 34.72 ± 0.18 |
| 17 | Ethyl Trans-2-Octenoate | — | — | — | 111.5 ± 2.30 |
| 18 | Chloroformic Acid N-Octyl Ester | — | — | 24.32 ± 0.22 | 76.52 ± 0.90 |
| 19 | Ethyl 3-Methylthiopropionate | — | — | 22.42 ± 1.14 | 55.92 ± 0.34 |
| 20 | Hexyl Hexanoate | — | 25.44 ± 2.68 | 131.26 ± 4.18 | — |
| 21 | Ethyl Caprate | — | — | — | 35.96 ± 3.78 |
| 22 | Ethyl 3-Hydroxyhexanoate | — | — | — | 50.94 ± 1.44 |
| 23 | Ethyl Phenylacetate | — | — | — | 59.28 ± 0.60 |
| 24 | Methyl (2Z,4E)-2,4-Decadienoate | — | — | — | 132.58 ± 0.96 |
| 25 | Phenethyl Acetate | — | — | 35 ± 1.40 | 74.4 ± 0.84 |
| 26 | Fema 3148 | — | — | 18.56 ± 0.40 | 457.04 ± 14.06 |

**Table 1.** *Cont.*

| No. | Compounds | 0 DAH (µg kg$^{-1}$) | 5 DAH (µg kg$^{-1}$) | 10 DAH (µg kg$^{-1}$) | 15 DAH (µg kg$^{-1}$) |
|---|---|---|---|---|---|
| 27 | Pentanoic Acid, 2,2,4-Trimethyl-3-Carboxyisopropyl, Isobutyl Ester | 10.72 ± 1.04 | — | — | 27.34 ± 2.68 |
| 28 | Phosphorochloridic Acid, Propyl Undecyl Ester | — | — | — | 70.46 ± 1.10 |
| 29 | Ethyl 9-Tetradecenoate | — | — | — | 74.4 ± 5.10 |
| 30 | Isophthalic Acid, Ethyl 2-Propylphenyl Ester | 9.68 ± 0.34 | — | — | — |
| 31 | Ethyl Palmitate | — | — | — | 90.64 ± 4.88 |
| | Ketones | | | | |
| 32 | 6-Methylhept-5-En-2-One | — | — | 43.18 ± 0.56 | 134.58 ± 0.90 |
| 33 | (E)-4-Oxohex-2-Enal | 80.00 ± 2.00 | 66.7 ± 2.70 | 15.38 ± 0.54 | — |
| 34 | 1,5,6,7-Tetrahydro-4H-Indol-4-One | — | 27.74 ± 2.74 | — | — |
| 35 | N-Cyclobutylidenehydroxylamine | 62.88 ± 3.32 | 42.80 ± 0.42 | — | — |
| | Terpenoids | | | | |
| 36 | Cis-B-Farnesene | — | — | 12.42 ± 0.48 | 23.7 ± 1.12 |
| 37 | (E)-3,7-Dimethylocta-1,3,6-Triene | — | — | 30.56 ± 1.88 | 39.06 ± 1.28 |
| 38 | Zingiberene | — | — | 26.76 ± 1.58 | — |
| 39 | 2,6-Dimethyl-6-(4-Methyl-3-Pentenyl)Bicyclo[3.1.1]Hept-2-Ene | — | — | — | 35.92 ± 3.02 |
| 40 | Farnesene | — | 534.42 ± 53.72 | 3606.7 ± 83.56 | 7756.26 ± 88.28 |
| 41 | (-)-B-Chamigrene | — | — | 72.8 ± 0.62 | — |
| 42 | 1,3-Cyclopentadiene, 1,3-Bis(1-Methylethyl)- | — | — | 33.82 ± 1.06 | — |
| 43 | (3Z,6E)-3,7,11-Trimethyldodeca-1,3,6,10-Tetraene | — | — | 29.12 ± 1.36 | 44.00 ± 2.1- |
| 44 | 2,5-Cyclohexadiene, 1,4-Diethyl-1,4-Dimethyl-1H-Indene, | — | — | — | 173.44 ± 0.6- |
| 45 | 2,3,3A,4-Tetrahydro-3,3A,6-Trimethyl-1-(1-Methylethyl)- | — | — | 17.8 ± 0.16 | — |
| 46 | Cyperene | — | — | — | 33.6 ± 1.1- |
| 47 | Guaiazulene | — | — | 56.2 ± 1.5- | 70.96 ± 0.18 |
| 48 | Isolongifolene, 9,10-Dehydro- | — | — | — | 28.04 ± 1.66 |
| 49 | .Beta.-Vatirenene | — | — | — | 28.46 ± 1.64 |
| 50 | Aromadendrene, Dehydro- | — | — | 18.7 ± 0.34 | 38.10 ± 1.68 |
| | Aldehydes | — | — | — | — |
| 51 | Hexanal | 619.12 ± 4.60 | 1563.84 ± 19.48 | 818.24 ± 14.06 | 742.60 ± 23.88 |
| 52 | 3-Hexenal | 86.72 ± 4.90 | — | — | — |
| 53 | Trans-2-Hexenal | 273.02 ± 14.34 | 1621.42 ± 39.26 | 1013.08 ± 11.68 | 704.04 ± 7.98 |
| 54 | (E,E)-2,4-Heptadienal | — | 9.08 ± 0.10 | — | — |
| | Alcohols | | | | |
| 55 | 1-Hexanol | — | — | — | 32.20 ± 0.80 |
| | Benzenes | | | | |
| 56 | Benzaldehyde | 20.02 ± 0.60 | 23.88 ± 0.54 | — | — |
| 57 | Benzeneethanamine, N-Butyl-.Beta.,4-Bis[(Trimethylsilyl)Oxy]- | — | 8.68 ± 0.02 | — | — |
| 58 | Silane, [[4-[1,2-Bis[(Trimethylsilyl)Oxy]Ethyl]-1,2-Phenylene]Bis(Oxy)]Bis[Trimethyl- | 8.68 ± 0.40 | 12.52 ± 0.74 | 16.74 ± 0.18 | — |
| 59 | Phenylacetaldehyde | 21.08 ± 0.60 | 14.98 ± 2.08 | 18.24 ± 1.64 | — |
| 60 | 1H-Benzocycloheptene, 2,4A,5,6,7,8,9,9A-Octahydro-3,5,5-Trimethyl-9-Methylene- | — | 11.96 ± 1.18 | — | 111.62 ± 1.48 |

**Table 1.** *Cont.*

| No. | Compounds | 0 DAH (µg kg$^{-1}$) | 5 DAH (µg kg$^{-1}$) | 10 DAH (µg kg$^{-1}$) | 15 DAH (µg kg$^{-1}$) |
|---|---|---|---|---|---|
| 61 | Phenylacetaldehyde | $47.12 \pm 1.8$ | $26.76 \pm 2.48$ | $53.34 \pm 1.00$ | $56.14 \pm 2.68$ |
| 62 | Hexestrol, O-Trifluoroacetyl- | — | $23.52 \pm 1.84$ | $91.3 \pm 2.66$ | — |
| 63 | 4-Tert-Butylphenol | — | — | $155.8 \pm 2.12$ | — |
| 64 | 4-Butyl-Benzonitrile | — | — | $17.34 \pm 0.62$ | — |
| 65 | 1,2,3,4-Tetrahydrophenanthren-9-Ol | — | $20.3 \pm 1.60$ | — | — |
| 66 | 2,4-Di-T-Butylphenol | $206.46 \pm 28.52$ | $106.88 \pm 5.76$ | $198.00 \pm 9.36$ | $151.7 \pm 12.82$ |
| | Others | | | | |
| 67 | Hexamethylcyclotrisiloxane | $9.30 \pm 0.58$ | $10.92 \pm 0.20$ | — | — |
| 68 | Octamethylcyclotetrasiloxane | — | $12.16 \pm 1.06$ | — | — |
| 69 | 4-[2-(5-Nitro-Furan-2-Yl)-Vinyl]-Quinolin-2-Ylamine | $12.36 \pm 0.38$ | — | — | — |
| 70 | Cyclohexene Oxide | $16.34 \pm 1.76$ | — | — | — |
| 71 | Methyl Cyclohexane | — | $16.18 \pm 0.98$ | — | — |
| 72 | Dodecamethylcyclohexasiloxane | $8.02 \pm 0.50$ | $9.68 \pm 1.12$ | — | — |
| 73 | Methylcyclopentane | — | — | $40.94 \pm 0.44$ | — |
| 74 | Tetradecamethyl Cycloheptasiloxane | $15.58 \pm 0.18$ | $25.68 \pm 0.44$ | $24.08 \pm 1.20$ | $49.28 \pm 2.40$ |
| 75 | Piperazine | $8.52 \pm 0.18$ | — | — | — |
| 76 | N-Heptadecane | — | $12.00 \pm 0.32$ | — | — |
| 77 | Ethyl 3-Methylsulfanylprop-2-Enoate | — | — | $36.24 \pm 0.18$ | $71.4 \pm 1.56$ |
| 78 | Tricyclo[3.1.0.0(2,4)]Hexane, 3,6-Diethyl-3,6-Dimethyl-, Trans- | — | — | $27.48 \pm 0.64$ | $48.68 \pm 0.20$ |
| 79 | (1S,5S)-9-Borabicyclo[3.3.1]Nonan-9-Ol | — | — | — | $48.26 \pm 2.30$ |

Note: The data in the table cites from [8].

In addition to the VOCs that can be directly perceived, there are bound aroma compounds that cannot be directly perceived in the fruit [41]. The bound aroma compounds bind to carbohydrates in the fruit through glycosidic bonds and release VOCs under enzyme, acid, or ultrasonic treatment [42]. The glycoside-bound aroma substances need to be eluted multiple times during enrichment to remove impurities [41]. The enriched aroma compounds bound to glycosides can be hydrolyzed into VOCs by enzymatic hydrolysis, acid hydrolysis, ultrasonic treatment and other methods, and then analyzed by conventional analytical methods for aroma compounds [43]. The hydrolysis of bound aroma compounds in fruits mostly occurs during fruit ripening. At this time, the activity of β-glucosidase in the fruits gradually increases, and the glycosidic bond of bound aroma compounds breaks to produce a perceptible aroma [42,43]. β-glucosidase can hydrolyze glycosidic bonds and release bound aroma substances. The hydrolysis of bound substances in apples contributes to the production of volatiles. In the 'Dangshan pear,' the level of expression of β-glucosidase decreased, and the production of aroma volatiles decreased with fruit ripening [44,45]. Currently, there are still few studies on bound aroma substances.

## 4. The Biosynthetic Pathway of VOCs in the 'Nanguo' Pear

The biosynthesis of aroma volatiles in fruit mostly occurs during the late stage of fruit development [17]. Fatty acids, amino acids, and carbohydrates in fruit serve as precursors of the aroma volatiles that form under the catalysis of various enzymes [3]. The aroma volatiles of fruits are secondary metabolites, and various precursors synthesize aroma volatiles through different pathways. Based on the types of precursors, the metabolic pathways of aroma biosynthesis are divided into the fatty acid metabolic pathway, amino acid metabolic pathway, terpenoid metabolic pathway, and carbohydrate metabolic pathway among others [3].

### 4.1. Fatty Acid Pathway

The primary aroma components of the 'Nanguo' pear are esters, which are biosynthesized by fatty acid metabolism (Figure 1) [17,38]. The β-oxidation of fatty acids is the primary biosynthetic process, which provides alcohols and acyl-CoA molecules to form esters [1]. Studies have found that the aroma volatiles in intact fruits are produced by β-oxidation. When plant tissue cells are destroyed, aroma volatiles are produced through the lipoxygenase (LOX) pathway [3]. However, some studies have shown that as the fruits mature, the membrane permeability increases, which increases the activity of the LOX pathway in the intact fruit. At this time, the LOX pathway can substitute for β-oxidation [3,46].

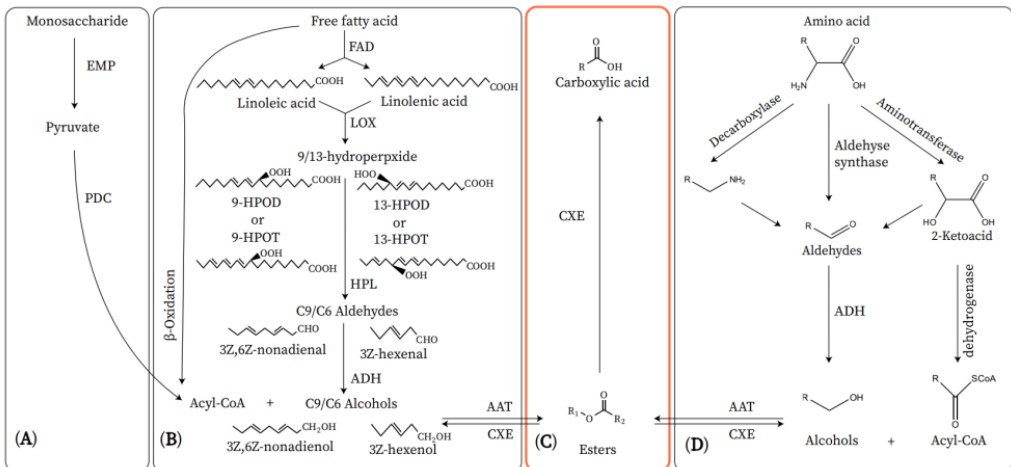

**Figure 1.** Metabolic pathway of volatile esters in fruit. (**A**) Monosaccharide pathway. (**B**) Fatty acid pathway. (**C**) CXE pathway. (**D**) Amino acid pathway. EMP: Embden-Meyerhof-Parnas; PDC: pyruvate dehydrogenase complex; FAD: fatty acid desaturase; LOX: lipoxygenase;9-HPOD: (10E,12Z)-9-hydroperoxy-10,12-oetadeeadienoic acid; 9-HPOT: (10E,12Z,15Z)-9-hydroperoxy-10,12,15-octadecatrienoic acid; 13-HPOD: (9Z,11E)-13-hydroperoxy-9,11-octadecadienoic acid, 13-HPOT: (9Z,11E,15Z)-13-hydroperoxy-9,11,15-octadecatrienoic acid; HPL: hydroperoxide lyase; ADH: alcohol dehydrogenase; AAT: alcohol acyl-CoA transferase; CXE: carboxylesterases.

In β-oxidation, acyl-CoA is reduced to aldehyde by acyl-CoA reductase, and the aldehyde is then reduced to alcohol by alcohol dehydrogenase (ADH) for alcohol acyltransferase (AAT) to produce esters [47]. The substrates of the LOX pathway are linolenic acid and linoleic acid, which can be obtained from free fatty acids under the action of fatty acid desaturase enzymes [37].

Linolenic acid and linoleic acid are derived through the LOX pathway into unsaturated short-chain alcohols, aldehydes, and esters [48]. Hydroperoxide lyase (HPL) is a downstream enzyme of LOX, which catalyzes the cleavage of hydroperoxide, the reaction product of LOX, into short-chain aldehydes [49]. Plant HPL is divided into two isozymes based on the difference of substrate peroxy group position. 13-HPL catalyzes the cleavage of the 13-position peroxy to produce $C_6$ compounds, while 9-HPL cleaves the 9-position peroxy to form $C_9$ compounds [50]. Next, alcohol dehydrogenase (ADH) catalyzes the interconversion of aldehydes and the corresponding alcohols. Finally, AAT catalyzes the reaction of acyl-CoA with alcohols to produce a variety of esters [47]. The alcohols involved in the reaction can be produced by the LOX pathway or reduced by short-chain acids produced by β-oxidation [1]. In addition, the LOX pathway can also produce jasmonic acid and its derivatives. In the allene oxide synthase (AOS) branch of the LOX pathway, 13-hydroxyperoxylinolenic acid is converted into 12,13-epoxyoctadecatrienoic acid through AOS, and jasmonic acid is then produced through a series of reactions. Jasmonic acid can be converted into the volatile ester methyl jasmonate by jasmonic acid carboxyl methyltransferase [51].

### 4.2. Amino Acid Pathway of Ester Biosynthesis

The amino acid metabolic pathway is also an important way to biosynthesize fruit aroma volatiles [52]. Aliphatic alcohols, aldehydes, and esters that contain branched chains can be biosynthesized through the amino acid metabolic pathway (Figure 1) [52]. A previous study found that the amino acids leucine, isoleucine, and valine could be the precursor of volatile alcohols, aldehydes, and esters in fruits, such as banana (*Musa* spp.), apple, strawberry (*Fragaria* × *ananassa*), and tomato (*Solanum lycopersicum*) [2,53,54]. In strawberries, alanine can also serve as the precursor for volatile ethyl esters, which can be produced by AAT [54]. Amino acids are converted to the corresponding $\alpha$-keto acids by aminotransferases, which are the key intermediates to convert amino acids into volatiles. $\alpha$-Keto acids are then converted to volatile aldehydes or acyl-CoA in the substrate of $\alpha$-keto acid decarboxylase or $\alpha$-keto dehydrogenase. Subsequently, aldehyde and acyl-CoA are converted to esters by ADH and AAT [55].

### 4.3. Carbohydrate Pathway

Carbohydrates are not only the energy source of fruit metabolism but also an important source of fruit flavor, which can act as precursors for the biosynthesis of aroma volatiles (Figure 1) [17]. Carbohydrates can be decomposed into pyruvate by the Embden-Meyerhof-Parnas (EMP) pathway, and acetyl-CoA can be produced under the action of pyruvate dehydrogenase complex (PDC), which can be involved in the fatty acid pathway and contribute to the formation of esters [1]. Another pathway is that pyruvate forms acetaldehyde under the action of PDC, then acetaldehyde is reduced to ethanol under the catalysis of ADH, and then the ester is synthesized, but the pathway has not been confirmed.

Terpenoid Pathway

Terpenoids are biosynthesized from acetyl-CoA and pyruvate provided by carbohydrates in plastids and the cytoplasm. Although fatty acid oxidation is one of the primary pathways for the production of acetyl-CoA, this process may have little to do with the formation of terpenoids because fatty acid oxidation occurs in the peroxisome [1]. Terpenoids are the most abundant secondary metabolites, which are the primary aroma volatiles of citrus and grapes (*Vitis vinifera*) [56,57]. The terpenoids in the 'Nanguo' pear are primarily derived from $\alpha$-farnesene [19]. $\alpha$-Farnesene is a sesquiterpene-like volatile that can be biosynthesized via the mevalonate pathway [52]. The MVA pathway is carried out in the cytoplasm MEP pathway is in the plastid [52]. The biosynthetic precursors of terpenoids are isopentenyl pyrophosphate (IPP) and dimethylallyl diphosphate (DMAPP) [52,58]. Its biosynthesis has two pathways, which include the 2-C-methyl-D-erythritol-4-phosphate (MEP) pathway and the mevalonate pathway (MVA) (Figure 2) [52]. The products of the MEP pathway are monoterpenes and diterpenes, and the products of the MVA pathway are sesquiterpenes and triterpenes [58]. Acetyl-CoA is catalyzed by an enzyme to produce isopentenyl pyrophosphate (IPP). IPP is catalyzed by IPP isomerase to produce DMAPP, and it is then catalyzed by an enzyme to produce geranyl pyrophosphate (GPP) and farnesyl pyrophosphate (FPP) [52]. FPP is catalyzed by an enzyme to synthesize $\alpha$-farnesene.

Moreover, the aroma of fruits is also regulated by carboxylesterases (CXE), which is an esterase that hydrolyzes esters [59]. Studies have found that the CXE in tomato and peach (*Prunus persica*) can use acetate as a substrate [60,61]. In pears, the content of CXE decreased with the extension of storage time, which could increase the accumulation of volatile esters in pears by reducing the degradation of esters [8].

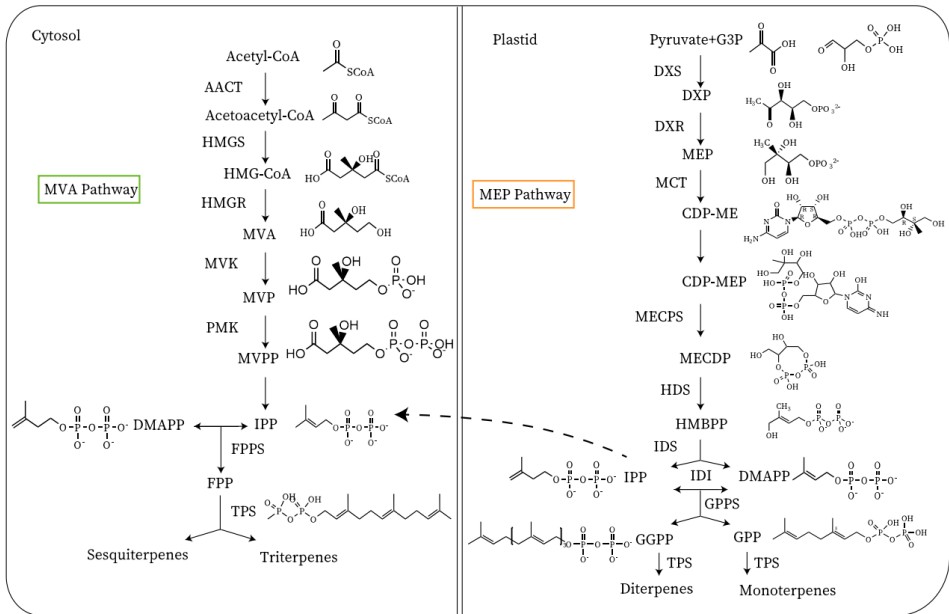

**Figure 2.** Synthesis of terpenoid volatile organic compounds. AACT, acetyl-CoA acetyl-transferase; HMG-CoA, hydroxymethylglutaryl-CoA; HMGS, HMG-CoA synthase; HMGR, hydroxymethylglutaryl-CoA; MVA, mevalonic acid; MVK, mevalonate kinase; MVP, mevalonate 5-phosphate; PMK, phosphomevalonate kinase; MVPP, phosphomevalonate kinase; FPP, farnesyl pyrophosphate; FPPS, FPP synthase; G3P, glyceraldehyde 3-phosphate; DXS, DXP synthase; DXP, 1-deoxy-d-xylulose 5-phosphate; DXR, 1-deoxy-d-xylulose 5-phosphate reductoisomerase; MEP, methylerythritol phosphate; CDP-ME, 4-diphosphocytidyl-2-C-methyl-d-erythritol; CDP-MEP, CDP-ME 2-phosphate; MECDP, 2-C-methyl-d-erythritol 2,4-cyclodiphosphate; MECPS, MECPD synthase; HDS, 4-hydroxy-3-methylbut-2-en-1-yl diphosphate synthase; HMBPP, (E)-4-hydroxy-3-methylbut-2-en-1-yl diphosphate; IDS, isopentenyl diphosphate synthase; IPP, isopentenyl pyrophosphate; IDI, isopentenyl pyrophosphate isomerase; DMAPP, dimethylallyl pyrophosphate; GGPP, geranylgeranyl pyrophosphate; GGPPS: GGPP synthase; TPS: terpene synthase; GPP, geranyl pyrophosphate; GPPS, GPP synthase; MCT, 2-C-methyl-d-erythritol 4-phosphate cytidylyltransferase.

## 5. Factors That Influence the Production of Aroma Volatiles in the 'Nanguo' Pear

Influencing factors such as cultivation and management conditions, fruit maturity, and post-harvest treatment can affect the aroma biosynthesis of fruits by affecting the state before, during, and after harvest (Figure 3). There is insufficient information on how these factors change the aroma metabolic pathway.

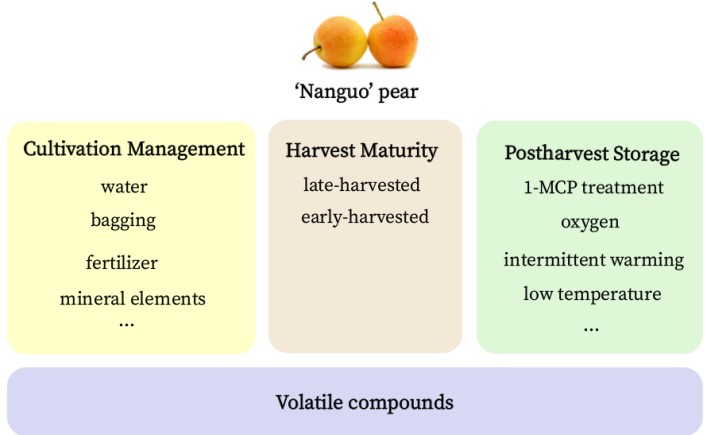

**Figure 3.** Factors affecting the production of volatile compounds in 'Nanguo' pear.

### 5.1. Conditions of Cultivation Management

Soil mineral elements, water, fertilizer and other factors can affect primary and secondary metabolism. In most cases, primary metabolism is first affected, which indirectly affects the formation of fruit aroma substances [62]. The results showed that the types and contents of aroma substances in 'Yali' pear fruits were significantly correlated with soil organic matter, nitrogen and zinc, and had high correlation with phosphorus, potassium and boron, and low correlation with iron [63]. Nitrogen is an indispensable element for fruit metabolism. Excessive nitrogen will affect the production of fruit aroma volatiles [64]. The content of butyl butyrate and hexyl butyrate will be significantly reduced by applying excessive nitrogen fertilizer to strawberries [65]. It is speculated that the metabolic pathway of butyric acid to the ester is inhibited [65]. Microbial fertilizer can increase the number of soil microorganisms, enrich soil nutrients and improve the soil environment. It was found that microbial fertilizer treatment significantly increased the content of soluble solids and volatile substances in fruits [66]. The mechanism was that microbial fertilizer increased the expression levels of *AAT1*, *AAT2*, *LOX*, *ADH* and *HPL* in fruits [67]. Water plays an important role in plant production, primary and secondary metabolism, so many authors have studied the effects of water limitation on aroma compounds, and these studies have focused on grapes [62]. The study found that reducing water and nitrogen supply before harvest can increase the VOCs content of grapes [68]. The pre-harvest bagging technology is to bag the fruit outside, and after a period of cultivation on the tree, the bag is removed or harvested together with the bag in the early stage of fruit harvest [5]. The content of total aroma substances and the production of ethyl butyrate and hexyl acetate decreased in the bagged 'Nanguo' pear [69]. By summarizing the effects of cultivation and management conditions on pear and other fruit aroma substances, it was found that appropriate soil, fertilizer and water management systems will help produce more valuable fruits.

### 5.2. Harvest Maturity

Maturity is one of the factors that affects the content of fruit VOCs. Recently, fruit growers often harvest early to corner the market in production. Although immature fruits are convenient for storage and transportation and bring considerable economic benefits, the quality of fruits harvested early is not the best, and there will be problems, such as poor color and aroma [70,71]. Pears produce different characteristic aroma compounds at different maturity stages. In the early stage of 'Nanguo' pear ripening, the content and species of aldehydes were the highest, and the content and species of esters increased with fruit ripening. In addition to the fruit aroma, the early harvested 'Nanguo' pears also mixed with some grass aroma due to its high hexanal content. The late-harvested 'Nanguo' pear had increased esters and showed strong fruit aroma [19]. The early harvested and late-harvested 'Nanguo' pears produce very different characteristic VOCs [19]. The early harvested 'Nanguo' pear is mixed with some grass aroma in addition to fruit aroma, and the late-harvested 'Nanguo' pear shows a strong fruit aroma [19].

### 5.3. Postharvest Storage Methods and Conditions

The optimal harvest time for the 'Nanguo' pear lasts for only 20 days [72]. The storage period of the 'Nanguo' pear can be effectively prolonged under low temperature conditions. However, long-term refrigeration leads to the deterioration of fruit flavor and aroma through the loss of compounds, such as volatile esters, which result in less valuable stored fruit [73,74]. Low temperature will reduce the respiratory rate and metabolic activity of fruits [72]. The biosynthesis of aroma volatiles requires the participation of many enzymes, which are decreased by low temperatures [3]. Refrigeration can reduce the activity of the key enzyme ATT and downregulate the expression of *AAT* in the biosynthetic pathway of ester aroma volatiles [38]. To explore the effect of temperature on aroma compounds, intermittent warming (IW) in the 'Nanguo' pear was implemented during refrigeration [16]. The results showed that the activity and gene expression of several key enzymes related to the biosynthesis of ester aroma compounds in the 'Nanguo' pear increased, which

promoted the biosynthesis of ester aroma compounds and maintained high aroma quality after refrigeration [38]. In addition, pre-storage at low temperatures can also maintain good aroma quality, since ethylene signal transduction was active, and the ester aromas were released in the 'Nanguo' pear [13,29,72]. There was a positive correlation between ethylene production and primary volatile esters, indicating that ethylene was a contributor to the induction of aroma-related esters in 'Nanguo' pears [73]. After low-temperature treatment, ethylene production was low, causing a loss of volatile esters [72]. In addition, the content of oxygen can also affect the biosynthesis of VOCs, and ultra-low oxygen storage conditions will inhibit the biosynthesis of pear fruit esters [74].

Some measures can be taken to prolong the storage time of fruit in production. 1-Methylcyclopropene (1-MCP) is an effective ethylene inhibitor, which can bind to the ethylene receptor first and inhibit the biosynthesis of endogenous ethylene and the induction of exogenous ethylene, thus, prolonging the storage time of fruits [75]. 1-MCP inhibited the effect of ethylene and the biosynthesis of aroma volatiles in the 'Nanguo' pear [72,73]. The activity of the limiting factor AAT in the ester biosynthetic pathway was regulated by ethylene. In apples, 1-MCP reduced the expression of *MdAAT1* and *MdAAT2* [76]. Therefore, the content of esters in the 'Nanguo' pear after 1-MCP treatment was greatly reduced [38]. Another study found that the application of 1-MCP could inhibit ADH activity and reduce the content of ethanol and acetaldehyde during storage [77].

## 6. Hormone Regulation of Aroma Volatiles in the 'Nanguo' Pear

Plant hormones play an important role during the entire process of plant growth and development and plant resistance and defense [78]. The regulation of plant hormones on the biosynthesis of fruit aroma volatiles is also a hot issue. The following sections will focus on the effects and related mechanisms of ethylene, methyl jasmonate (MeJA), salicylic acid and glycine betaine on the biosynthesis of VOCs of fruit.

### 6.1. Ethylene

Ethylene is a plant hormone that is closely related to plant maturation and senescence [79]. Fruits are usually divided into climacteric fruit and non-climacteric fruit [80]. The difference between the two is that the former will produce peaks of respiration and ethylene release near fruit ripening. The 'Nanguo' pear is a respiratory climacteric fruit [8,80]. The peaks of aroma and ethylene release of the 'Nanguo' pear appeared almost simultaneously [73]. Esters are the primary types of aroma volatiles in 'Nanguo' pears [8]. Ethylene was related to the biosynthesis of aroma VOCs in 'Nanguo' pears. A linear regression analysis showed that the production of ester aromas in a 'Nanguo' pear positively correlated with the production of ethylene [74,81]. The contents of ethyl butyrate and ethyl caproate in 'Nanguo' pears treated with exogenous ethylene were significantly higher than those in the control fruit [81]. The correspondence between ethylene content and volatile esters also exists in other fruits, such as peach and kiwifruit (*Actinidia chinensis* Planch. var *chinensis*) [82,83].

Ethylene biosynthesis includes two key steps. S-adenosylmethionine (SAM) is converted to 1-aminocyclopropane-1-carboxylic acid (ACC) by ACC synthase (ACS), and ethylene is then formed from ACC by ACC oxidase (ACO) [84]. The ethylene signaling pathway involves the five major components ethylene receptors ETR1, ERS1, ETR2, EIN4, and ERS2; CTR1, a negative regulator; EIN2, a positive regulatory factor in the endoplasmic reticulum (ER); EIN3, a transcription factor located in the nucleus; and the transcription factor ERF [85,86]. *PuERS1*, *PuEIN4*, *PuEIN2*, *PuERF* and *PuEIN3* are the five key genes involved in ethylene signal transduction in the 'Nanguo' pear [73]. Treatment with exogenous ethylene increased their expression and promoted the production of endogenous ethylene, which eventually increased the number and content of volatile esters in the fruit [74].

The treatment of 'Nanguo' pears with exogenous ethylene can increase the activities of LOX, ADH, AAT and other key enzymes; thus, promoting the biosynthesis of esters and

reducing the loss of aromas caused by refrigeration [73]. Dof (DNA-binding with one finger) proteins recognize the binding of AAAG sequences to the promoters of their target genes and play key roles in a variety of physiological processes in higher plants [87]. Studies have found that the activity of AAT in apples is highly regulated by ethylene [88]. The possible mechanism that involves the promotion of AAT biosynthesis by exogenous ethylene in the 'Nanguo' pear is to enhance the expression of *PuERF13* and *PuDof2.5*; *PuDof2.5* binds to the *PuAAT1* promoter and upregulate its expression [81]. *PuERF13* promotes the transcriptional regulation of *PuDof2.5* by interacting with *PuDof2.5* (Figure 4). In a peach, ethylene binds to the *PpLOX4* promoter through a complex formed by transcription factors PpERF5 and PpERF7, activates its expression, and promotes the biosynthesis of aroma during peach fruit ripening [89]. With the maturation of an apple, the activities of *MdADH1* and *MdADH2* gradually decreased, and the activity of *MdADH3* increased. In the 'Nanguo' pear, *PuADH1* activity increased with fruit ripening. *MdADH3* and *PuADH1* may be regulated by ethylene, which may affect the flavor characteristics by limiting the supply of alcohols produced by esters during ripening [8,90]. The molecular mechanism used by exogenous ethylene to regulate ADH remains unknown [81].

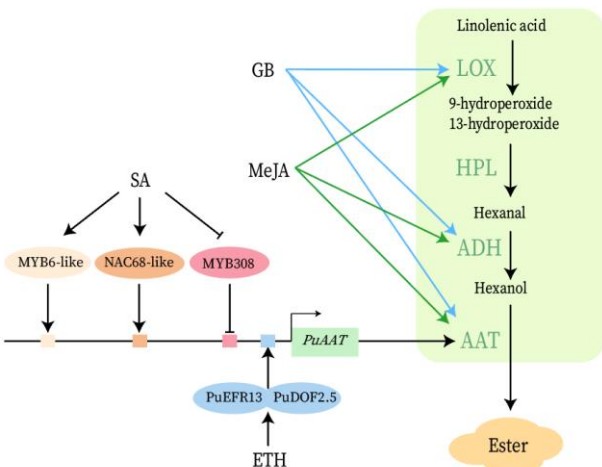

**Figure 4.** Phytohormones regulate ester biosynthesis in the 'Nanguo' pear. GB, Glycine betaine; MeJA, Methyl jasmonate; SA, Salicylic Acid; ETH, ethylene. LOX: lipoxygenase; HPL: hydroperoxide lyase; ADH: alcohol dehydrogenase; AAT: alcohol acyl-CoA transferase.

### 6.2. Methyl Jasmonate

MeJA is a volatile derivative of jasmonic acid, a plant hormone associated with plant stress resistance [91]. When plants are subjected to stress, jasmonic acid and MeJA can regulate defense responses by accumulating antioxidant enzyme activity [92]. It has been reported that MeJA can promote changes in gene expression responsible for fruit aroma, particularly volatile $C_6$ compounds, including $C_6$ aldehydes, alcohols and esters with six carbons, which play an important role in defense mechanisms and have a fresh green aroma [93,94]. Jasmonic acid is usually biosynthesized by the fatty acid metabolic pathway. Linoleic acid is catalyzed by LOX to produce hydroperoxide, which is then β-oxidized to form jasmonic acid. Jasmonic acid can be methylated to form MeJA [51,95]. Previous studies have shown that the aroma of 'Nanguo' pear fruit often decreases after refrigeration [38,74]. Under MeJA treatment, the contents of ethyl butyrate, ethyl caproate and hexyl butyrate increased [29]. Ethylene substantially contributes to the production of fruit aromas [96]. On the one hand, MeJA can increase the contents of linoleic acid and linolenic acid, the primary precursors involved in the biosynthesis of aroma volatiles in the fruit, and promote the biosynthesis of esters in the 'Nanguo' pear [29]. Alternatively, MeJA can promote the expression of the genes *PuAAT*, *PuADH3*, *PuADH5*, *PuADH9*, *PuLOX1* and *PuLOX3* that are related to the biosynthesis of esters to increase the activity of AAT, ADH and LOX to promote the biosynthesis of 'Nanguo' pear esters [29]. In addition, MeJA is a signaling

molecule that regulates ethylene production and its signal transduction pathways [29]. The activities of ACS and ACO and the levels of expression of *PuACS1* and *PuACO1* increased in 'Nanguo' pears treated with MeJA, which contributed to the accumulation of ethylene and the release of aroma volatiles (Figure 4). The upregulated expression of *PuERS1*, *PuETR1*, *PuEIN4*, *PuEIN2* and *PuERF1* was also related to the higher content of aroma-related esters produced by 'Nanguo' pears after MeJA treatment [97]. This is consistent with the results in peaches. MeJA treatment alleviated the loss of lactones by inducing the expression of *PpACS3*, *PpACS4* and *PpACO* involved in ethylene biosynthesis in peaches [98]. However, the specific regulatory mechanism of MeJA involved in ethylene signal transduction merit further exploration.

### 6.3. Salicylic Acid

SA is a plant hormone associated with fruit ripening and a signaling molecule involved in plant defense [99,100]. SA can increase the biosynthesis of VOCs, particularly esters, in the 'Nanguo' pear [29]. The activities of LOX, ADH and ATT were activated, and the levels of expression of *LOX*, *ADH* and *ATT* increased, indicating that SA enhances the aroma of the 'Nanguo' pear by promoting the activity and expression of key enzymes in the biosynthesis of esters [29]. AAT is a key enzyme in the final step of ester aroma biosynthesis, which is regulated by ethylene [6]. Higher AAT activity results in higher contents of esters [52,88]. It was found that SA could promote the expression of PuMYB308-like and PuMYB6-like, and the two transcription factors activated the expression of *PuAAT* by directly binding to its promoter. At the same time, SA treatment inhibited the negative regulation of PuNAC68-like on *PuAAT*, which was conducive to the formation of esters [12]. SA can positively regulate the *PuAAT* gene and promote the expression of transcription factors, inhibit the negative regulation of some transcription factors on *PuAAT* and thus, affect the biosynthesis of aromas in the 'Nanguo' pear (Figure 4) [12]. SA can delay fruit senescence by delaying ethylene production and slowing down the respiration rate of fruits, and can be used as a preservative in production [101]. MeJA and exogenous ethylene treatment will promote the production of ethylene to a certain extent, which may slow down the shelf life of fruit. Studies have shown that methyl salicylate treatment maintains the content of flavor-related volatiles in tomato fruit after cold storage [102]. SA and its derivatives may become an excellent preservative that has no significant effect on fruit aroma quality.

### 6.4. Glycine Betaine

Glycine betaine (GB) is an alkaloid that can maintain the balance of intracellular osmotic pressure when plants are subjected to stress and serve as an osmotic regulator [103]. GB has a positive regulatory function in enhancing the cold tolerance of sweet cherry (*Prunus avium* L.) [104]. The contents and types of esters, linoleic acid and linolenic acid and the activities of LOX, ADH and AAT in 'Nanguo' pears increased, and the expression levels of *PuLOX1*, *PuLOX2*, *PuADH3*, *PuADH4* and *PuADH9* increased after exogenous treatment with GB (Figure 4) [105]. The ethylene production of 'Nanguo' pears treated with GB was higher than that of the control. The expression levels of *PuACO1*, *PuACO2* and *PuACS* directly related to ethylene biosynthesis and ethylene response factors *PuERF2*, *PuERF109-like*, *PuERF071-like*, *PuERF015*, *PuERF114*, *PuRAV1-like* and *PuERF113-like* were up-regulated [105]. GB directly promoted the biosynthesis of ester aroma volatiles in 'Nanguo' pears by increasing the contents of ester aroma precursors and key enzymes in the biosynthetic pathway [106]. GB treatment was found to increase the contents of the key enzymes betaine dehydrogenase (BADH) and choline monooxygenase (CMO) in GB biosynthesis and promoted its biosynthesis, while a high content of GB promoted the biosynthesis and signal transduction of ethylene, thus, indirectly promoting the production of ester aromas [105].

The hormones described above can regulate the biosynthesis of aroma substances in 'Nanguo' pear in a variety of ways. MeJA, salicylic acid, and glycine betaine can promote

the production of ethylene and stimulate ethylene signal transduction [12,29]. Ethylene plays an important role in the biosynthesis of fruit aromas, and LOX, ADH, AAT and other key enzymes in the aroma biosynthetic pathway are regulated by various plant hormones [12,29,83]. In addition to the effects on the biosynthesis of fruit aroma volatiles from these four hormones, many hormones also affect the biosynthesis of aroma volatiles. Melatonin is an example. This compound is an indole tryptamine, which is widely found in plants and animals, and can delay plant senescence and maintain fruit quality [107,108]. After treatment with melatonin, the release of ethylene and its peak decreased in pears, and the peak of LOX enzyme activity was delayed, which affected the release of aromas of 'Nanguo' pears [109,110].

## 7. Conclusions and Future Perspectives

The production of volatiles is affected by many factors. So far, our understanding of how these factors interact is limited. Currently, the contribution of volatiles to aroma is evaluated by OAV, but the matrix used in OAV calculation is mostly water, which is different from the real matrix of aroma, and the contribution to aroma cannot be well expressed [111].

There are many studies on bound aroma substances in grapes and strawberry fruits, but few in apples, pears and other fruits [41,112]. The contribution of bound aroma substances to aroma is not clear. Although the characteristic volatile compounds in 'Nanguo' pears have been identified and quantified, the research on the biosynthetic pathway and precursor verification of their volatiles is still limited. For example, the biosynthetic pathway of carbohydrates in ester aroma volatiles is still unclear. At the late ripening stage of the 'Nanguo' pear, the content of ethanol and acetaldehyde in the fruit increased rapidly [79]. Whether these ethanols will participate in the biosynthesis of esters as precursors is unknown. Most studies have focused on the biosynthesis of volatile aroma, and few researchers have studied the hydrolysis of esters. CXE is an esterase, but its substrate in 'Nanguo' pear is not clear. It was found that 1-MCP treatment significantly inhibited ADH activity, indicating that ADH may be regulated by ethylene, but its molecular mechanism is not clear, and more research is needed from the translation and post-translation levels [79]. When using plant hormones to treat the 'Nanguo' pear, it was found that ethylene seemed to be an important participant in regulating the biosynthesis of fruit aroma, but its specific molecular mechanism was not clear enough. In the future, a plant hormone crosstalk network with ethylene as the core may be constructed.

Furthermore, with the development of molecular biology, important aspects of the posttranslational modifications and epigenetics are exciting directions in the field of study of fruit aromas.

**Author Contributions:** Writing—original draft preparation, Z.Z.; writing—review and editing, Z.Y. All authors have read and agreed to the published version of the manuscript.

**Funding:** This research was funded by Natural Science Foundation of Liaoning Province, grant number 2022-MS-261, the Research Foundation of Education Bureau of Liaoning Province, grant number LJKMZ20221025, General Higher Education Undergraduate Teaching Reform Research Project of Liaoning Province, grant number 2022-444, Postgraduate Education and Teaching Research Project of Shenyang Agricultural University, grant number 2022-yjs-39, and National Natural Science Foundation of China grant number 31801848.

**Data Availability Statement:** Not applicable.

**Acknowledgments:** We also thank to LiYong Qi (Shenyang Agricultural University) for providing helpful comments on the manuscript.

**Conflicts of Interest:** The authors declare that they have no competing interest.

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
