# Peer review of "The Aroma Volatile in ‘Nanguo’ Pear: A Review"

_horticulturae, doi:10.3390/horticulturae9030339_

Round 1

Reviewer 1 Report

Review report for  a review manuscript 

In general, This Review is very interesting. It scientifically sounds with a great topic and a great impact on the field. Although, this manuscript seems to be short as a review article But it is Rey organized and specific to the topic. I believe that it will be suitable for publication after a minor revision.

Detailed comments:

-The English language and /writing style is fine needs some moderate  check spelling and moderate grammar check.

Abstract

_This section is well written and the  aim of the review was clearly stated . 

Keywords:

-The keywords has been chosen very carefully and accurately.

  • 1. INTRODUCTION 

-This section is very short and doesn’t provide enough background. It needs to be elongated and improved.

5.Influencing Factors of Aroma Volatile in Nanguon pear

-This topic doesn’t provide sufficient background and it is missing enough relevant references

-This section needs to be elongated and enriched with more background about this topic.

-The author is advised to creat an illustrating diagram for this topic.

  1. Hormone Regulation of Aroma Volatile in Nanguon pear

-This topic doesn’t provide sufficient background and it is missing enough relevant references

-This section needs to be elongated and enriched with more background about this topic.

  • Also the author is advised to creat an illustrating diagram for this topic.

References

This section is well written and UpToDate. Except for the mentioned topics that needs more background and related citations were not enough .

Author Response

Review 1

Comment 1: Review report for a review manuscript

In general, This Review is very interesting. It scientifically sounds with a great topic and a great impact on the field. Although, this manuscript seems to be short as a review article. But it is Rey organized and specific to the topic. I believe that it will be suitable for publication after a minor revision.

Response: Thank you for your positive comments, we have added some content according to your request.

Comment 2: Detailed comments:

-The English language and /writing style is fine needs some moderate check spelling and moderate grammar check.

Response: Thank you very much for your suggestion. Now we’ve thoroughly corrected the language by English native speaker in this revised version and we hope that you are satisfied with this version.

Comment 3: Abstract

_This section is well written and the aim of the review was clearly stated

Response: Thank you for your comments. We appreciate your revision of the manuscript.

Comment 4: Keywords:

-The keywords has been chosen very carefully and accurately.

Response: Thank you, we really appreciate your review and kind correction of our manuscripts.

Comment 5: 1. INTRODUCTION

-This section is very short and doesn’t provide enough background. It needs to be elongated and improved.

Response: Thanks for your comment. We have modified this part.

Line24-27: ‘Plants have the ability to synthesize, accumulate and release volatiles [1]. For plants, the release of volatile substances may be to spread seeds. For humans, these volatile compounds (VOCs) may have antibacterial or anticancer activity. People will also extract some of these ingredients to make perfume [2].’

Line40-46: ‘Considering the important influence of VOCs on flavor, a lot of research has been carried out on the VOCs of fruits. Breakthroughs have been made in the biosynthesis path of fruit aroma substances, the identification of key structural genes, and the regulation of exogenous hormones on the metabolism of aroma substances [1,11,12]. With the emergence of precision instruments such as gas chromatography-mass spectrometry (GC-MS), the research on aroma volatiles has become more in-depth, and the method for determining aroma volatiles has gradually matured.’

Line59-62: ‘Nanguo’ pear is a good example to study fruit aroma. So far, 79 volatile chemicals have been observed in ‘Nanguo’ pear. These compounds include esters, alcohols, ketones, terpenes, aldehydes and benzene compounds [8]. The types and concentrations of these volatiles vary according to their cultivation conditions and maturity [15,19].’

Comment 6: 5. Influencing Factors of Aroma Volatile in Nanguon pear

-This topic doesn’t provide sufficient background and it is missing enough relevant references

-This section needs to be elongated and enriched with more background about this topic.

Response: Thanks. We have modified the discussion part to make it concise and clear as your suggested.

Line356-379: ‘Soil mineral elements, water, fertilizer and other factors can affect primary and secondary metabolism. In most cases, primary metabolism is first affected, which indirectly affects the formation of fruit aroma substances [63]. The results showed that the types and contents of aroma substances in ‘Yali’ pear fruits were significantly correlated with soil organic matter, nitrogen and zinc, and had high correlation with phosphorus, potassium and boron, and low correlation with iron [64]. Nitrogen is an indispensable element for fruit metabolism. Excessive nitrogen will affect the production of fruit aroma volatiles [65]. The content of butyl butyrate and hexyl butyrate will be significantly reduced by applying excessive nitrogen fertilizer to strawberries [66]. It is speculated that the metabolic pathway of butyric acid to ester is inhibited [66]. Microbial fertilizer can increase the number of soil microorganisms, enrich soil nutrients and improve soil environment. It was found that microbial fertilizer treatment significantly increased the content of soluble solids and volatile substances in fruits [67]. The mechanism was that microbial fertilizer increased the expression levels of AAT1, AAT2, LOX, ADH and HPL in fruits [68]. Water plays an important role in plant production, primary and secondary metabolism, so many authors have studied the effects of water limitation on aroma compounds, and these studies have focused on grapes [63]. The study found that reducing water and nitrogen supply before harvest can increase the VOCs content of grapes [69]. The pre-harvest bagging technology is to bag the fruit outside, and after a period of cultivation on the tree, the bag is removed or harvested together with the bag in the early stage of fruit harvest [5]. The content of total aroma substances and the production of ethyl butyrate and hexyl acetate decreased in bagged ‘Nanguo’ pear [70]. By summarizing the effects of cultivation and management conditions on pear and other fruit aroma substances, it was found that appropriate soil, fertilizer and water management systems will help produce more valuable fruits.’

Line395-401: ‘Pears produce different characteristic aroma compounds at different maturity stages. In the early stage of ‘Nanguo’ pear ripening, the content and species of aldehydes were the most, and the content and species of esters increased with fruit ripening. In addition to the fruit aroma, the early-harvested 'Nanguo' pears also mixed with some grass aroma due to its high hexanal content. The late-harvested 'Nanguo' pear had increased esters and showed strong fruit aroma [19].’

Line423-426: ‘There was a positive correlation between ethylene production and primary volatile esters, indicating that ethylene was a contributor to the induction of aroma-related esters in 'Nanguo' pear [74]. After low temperature treatment, ethylene production is low, causing loss of volatile esters [73].’

Comment 7: -The author is advised to creat an illustrating diagram for this topic.

Response: Thank you for your suggestion, we added ‘Figure 3. Factors affecting the production of volatile compounds in ‘Nanguo’ pear.’ In the manuscript.

Comment 8: Hormone Regulation of Aroma Volatile in Nanguon pear

-This topic doesn’t provide sufficient background and it is missing enough relevant references

-This section needs to be elongated and enriched with more background about this topic.

Response: Thank for the suggestion of improvement. We have rewritten the relevant content.

Line458-460: The correspondence between ethylene content and volatile esters also exists in other fruits, such as peach and kiwifruit (Actinidia chinensis Planch. var chinensis) [84,85].

Line482-489: In peach, ethylene binds to the PpLOX4 promoter through a complex formed by transcription factors PpERF5 and PpERF7, activates its expression, and promotes the biosynthesis of aroma during peach fruit ripening [91]. With the maturation of apple, the activities of MdADH1 and MdADH2 gradually decreased, and the activity of MdADH3 increased. In Nanguo pear, PuADH1 activity increased with fruit ripening. MdADH3 and PuADH1 may be regulated by ethylene, which may affect the flavor characteristics by limiting the supply of alcohols produced by esters during ripening [8,92].

Line500-503: It has been reported that MeJA can promote changes in gene expression responsible for fruit aroma, particularly volatile C6 compounds, including C6 aldehydes, alcohols and esters with six carbons, which play an important role in defense mechanisms and have a fresh green aroma [95,96].

Line523-525: This is consistent with the results in peach. MeJA treatment alleviated the loss of lactones by inducing the expression of PpACS3, PpACS4 and PpACO involved in ethylene biosynthesis in peach [100].

Line537-540: It was found that SA could promote the expression of PuMYB308-like and PuMYB6-like, and the two transcription factors activated the expression of PuAAT by directly binding to its promoter. At the same time, SA treatment inhibited the negative regulation of PuNAC68-like on PuAAT, which was conducive to the formation of esters [12].

Line545-551: SA can delay fruit senescence by delaying ethylene production and slowing down the respiration rate of fruits, and can be used as a preservative in production [103]. MeJA and exogenous ethylene treatment will promote the production of ethylene to a certain extent, which may slow down the shelf life of fruit. Studies have shown that methyl salicylate treatment maintains the content of flavor-related volatiles in tomato fruit after cold storage [104]. SA and its derivatives may become an excellent preservative that has no significant effect on fruit aroma quality.

Line555-563: GB has a positive regulatory function in enhancing the cold tolerance of sweet cherry (Prunus avium L.) [106]. The contents and types of esters, linoleic acid and linolenic acid and the activities of LOX, ADH and AAT in 'Nanguo' pear increased, and the expression levels of PuLOX1, PuLOX2, PuADH3, PuADH4 and PuADH9 increased after exogenous treatment with GB [107]. The ethylene production of 'Nanguo' pear treated with GB was higher than that of the control. The expression levels of PuACO1, PuACO2 and PuACS directly related to ethylene biosynthesis and ethylene response factors PuERF2, PuERF109-like, PuERF071-like, PuERF015, PuERF114, PuRAV1-like and PuERF113-like were up-regulated (Figure 4.) [107].

Comment 9: Also the author is advised to creat an illustrating diagram for this topic.

Response: Thank you for your suggestion, we added ‘Figure 4. Phytohormones regulate ester biosynthesis in ‘Nanguo’ pear.’ In the manuscript.

Comment 10: References

This section is well written and UpToDate. Except for the mentioned topics that needs more background and related citations were not enough.

Response: Thanks for your suggestions. We have added and cited relevant literature in this revised version.

Reviewer 2 Report

This report “Advances in aroma volatile research of ‘Nanguo’ pear” comprehensively reviews the study provides updated information on the extraction and identification methods, the species and composition, the biosynthesis pathway, the factors influencing and hormone regulation of aroma volatile in ‘Nanguo’ pear. However, this review lacks focus and rigor. The title of the review is too ambiguous and vague. Advances in what sense? The title is not fully expressing the text. Keywords are present in the title (Aroma volatiles, ‘Nanguo’ pear), choose other indexing terms for the article; The authors just presented figures that summarize the metabolic pathway of volatile esters in fruit and synthesis of terpenoid volatile organic compounds, the review deserves more figures and some tables, presenting details and results in the other topics addressed. A point that must imperatively be addressed in an isolated topic, is it about the functionality and applicability of the volatile compounds presented in the review, as far as applied science is concerned. The authors should critically discuss the existing literature, point out the knowledge gaps, and suggest further research. The manuscript slightly lacks coherence in storyline, and English language also needs careful editing for better readability. The review is poorly written hence, needs rewriting. The manuscript should be further strengthened by adding some more relevant papers. The literature search is insufficient, only few related research papers in the past five years are cited (41,3%, approximately), add the latest research results appropriately.

Author Response

Review 2

Comment 1: This report “Advances in aroma volatile research of ‘Nanguo’ pear” comprehensively reviews the study provides updated information on the extraction and identification methods, the species and composition, the biosynthesis pathway, the factors influencing and hormone regulation of aroma volatile in ‘Nanguo’ pear. However, this review lacks focus and rigor. The title of the review is too ambiguous and vague. Advances in what sense? The title is not fully expressing the text.

Response: Thanks for your advice. We have modified the title to ‘The aroma volatile in ‘Nanguo’ pear: a review.’

Comment 2: Keywords are present in the title (Aroma volatiles, ‘Nanguo’ pear), choose other indexing terms for the article; The authors just presented figures that summarize the metabolic pathway of volatile esters in fruit and synthesis of terpenoid volatile organic compounds, the review deserves more figures and some tables, presenting details and results in the other topics addressed. A point that must imperatively be addressed in an isolated topic, is it about the functionality and applicability of the volatile compounds presented in the review, as far as applied science is concerned.

Response: Thank you for your suggestion, we added ‘Table 1. Volatile compounds detected in ‘Nanguo’ Pear fruit during fruit ripening.’, ‘Figure 3. Factors affecting the production of volatile compounds in ‘Nanguo’ pear.’ And ‘Figure 4. Phytohormones regulate ester biosynthesis in ‘Nanguo’ pear.’ In the manuscript. We have corrected in the manuscript as your suggested.

Line24-27: ‘Plants have the ability to synthesize, accumulate and release volatiles [1]. For plants, the release of volatile substances may be to spread seeds. For humans, these volatile compounds (VOCs) may have antibacterial or anticancer activity. People will also extract some of these ingredients to make perfume [2].’

Line40-46: ‘Considering the important influence of VOCs on flavor, a lot of research has been carried out on the VOCs of fruits. Breakthroughs have been made in the biosynthesis path of fruit aroma substances, the identification of key structural genes, and the regulation of exogenous hormones on the metabolism of aroma substances [1,11,12]. With the emergence of precision instruments such as gas chromatography-mass spectrometry (GC-MS), the research on aroma volatiles has become more in-depth, and the method for determining aroma volatiles has gradually matured.’

Line59-62: ‘Nanguo’ pear is a good example to study fruit aroma. So far, 79 volatile chemicals have been observed in ‘Nanguo’ pear. These compounds include esters, alcohols, ketones, terpenes, aldehydes and benzene compounds [8]. The types and concentrations of these volatiles vary according to their cultivation conditions and maturity [15,19].’

Comment 3: The authors should critically discuss the existing literature, point out the knowledge gaps, and suggest further research.

Response: Thanks for your comment. We have rewritten the ‘Conclusions and Future Perspectives’ section.

Line587-608: ‘The production of volatiles is affected by many factors. So far, our understanding of how these factors interact is limited. Currently, the contribution of volatiles to aroma is evaluated by OAV, but the matrix used in OAV calculation is mostly water, which is different from the real matrix of aroma, and the contribution to aroma cannot be well expressed [113].

There are many studies on bound aroma substances in grape and strawberry fruits, but few in apple, pear and other fruits [41,114]. The contribution of bound aroma substances to aroma is not clear. Although the characteristic volatile compounds in ‘Nanguo’ pear have been identified and quantified, the research on the biosynthetic pathway and precursor verification of their volatiles is still limited. For example, the biosynthetic pathway of carbohydrates in ester aroma volatiles is still unclear. At the late ripening stage of ‘Nanguo’ pear, the content of ethanol and acetaldehyde in the fruit increased rapidly [79]. Whether these ethanols will participate in the biosynthesis of esters as precursors is unknown. Most studies have focused on the biosynthesis of volatile aroma, and few researchers have studied the hydrolysis of esters. CXE is an esterase, but its substrate in ‘Nanguo’ pear is not clear. It was found that 1-MCP treatment significantly inhibited ADH activity, indicating that ADH may be regulated by ethylene, but its molecular mechanism is not clear, and more research is needed from the translation and post-translation levels [79]. When using plant hormones to treat ‘Nanguo’ pear, it was found that ethylene seemed to be an important participant in regulating the biosynthesis of fruit aroma, but its specific molecular mechanism was not clear enough. In the future, a plant hormone crosstalk network with ethylene as the core may be constructed.’

Comment 4: The manuscript slightly lacks coherence in storyline, and English language also needs careful editing for better readability.

Response: Thank you for your comments. We have made a thorough revision of the language through native English speakers and hope that you will be satisfied with this version.

Comment 5: The review is poorly written hence, needs rewriting. The manuscript should be further strengthened by adding some more relevant papers.

Response: Thanks for your comments. Now we’ve thoroughly corrected the language by English native speaker in this revised version and we hope that you are satisfied with this version. We have added 11 latest articles in this revision.

Comment 6: The literature search is insufficient, only few related research papers in the past five years are cited (41,3%, approximately), add the latest research results appropriately.

Response: Thank you for your advice. We have added and cited 18 relevant articles in this revised edition, of which 11 are the latest.

Reviewer 3 Report

Manuscript is well prepared, is informative, and summarizes data on volatile compounds in specific pear cultivar. I propose some minor corrects:

Lines 73 and 83 - please use only surname,

Lines 99-101 - please correct data for this sentence "Using this method, we have found seventy-nine VOCs in ‘Nanguo’ pear, including four aldehydes, thirty-one esters, four ketones, one alcohol, fifty terpenoids, eleven benzenes, thirteen other substances",

Please be consequent - line 113 "nearly one hundred" vs. line 109 "we have found seventy-nine"; similarly line 109 "we" vs. 101 "they",

Can you add same data on "odor activity values (OAV)" of specific aromas of ‘Nanguo’ pear?

Figure 1 - please add full name of CXE pathway,

In paragraph "Conditions of Cultivation Management" please add same details on correlation between soil minerals/water management and specific pear aromas.

Line 479 - please use small letters 

Author Response

Review 3

Manuscript is well prepared, is informative, and summarizes data on volatile compounds in specific pear cultivar. I propose some minor corrects:

Comment 1: Lines 73 and 83 - please use only surname,

Response: Thank you for your advice. We have corrected it in the manuscript as you suggest.

Comment 2: Lines 99-101 - please correct data for this sentence "Using this method, we have found seventy-nine VOCs in ‘Nanguo’ pear, including four aldehydes, thirty-one esters, four ketones, one alcohol, fifty terpenoids, eleven benzenes, thirteen other substances",

Response: Thanks for your suggestions. We have revised the manuscript as you suggested.

Comment 3: Please be consequent - line 113 "nearly one hundred" vs. line 109 "we have found seventy-nine"; similarly line 109 "we" vs. 101 "they",

Response: Thank you for your comments. We modified it in the manuscript.

Comment 4: Can you add same data on "odor activity values (OAV)" of specific aromas of ‘Nanguo’ pear?

Response: Thanks, we really appreciate your scrupulous review and kind corrections to our manuscript. We added ‘Table 1. Volatile compounds detected in ‘Nanguo’ Pear fruit during fruit ripening’ in the manuscript.

Comment 5: Figure 1 - please add full name of CXE pathway,

Response: Thanks for your reminding. We have revised the manuscript according to your suggestions.

Comment 6: In paragraph "Conditions of Cultivation Management" please add same details on correlation between soil minerals/water management and specific pear aromas.

Response: Thanks. We have modified the discussion part to make it concise and clear as your suggested.

Line356-379: ‘Soil mineral elements, water, fertilizer and other factors can affect primary and secondary metabolism. In most cases, primary metabolism is first affected, which indirectly affects the formation of fruit aroma substances [63]. The results showed that the types and contents of aroma substances in ‘Yali’ pear fruits were significantly correlated with soil organic matter, nitrogen and zinc, and had high correlation with phosphorus, potassium and boron, and low correlation with iron [64]. Nitrogen is an indispensable element for fruit metabolism. Excessive nitrogen will affect the production of fruit aroma volatiles [65]. The content of butyl butyrate and hexyl butyrate will be significantly reduced by applying excessive nitrogen fertilizer to strawberries [66]. It is speculated that the metabolic pathway of butyric acid to ester is inhibited [66]. Microbial fertilizer can increase the number of soil microorganisms, enrich soil nutrients and improve soil environment. It was found that microbial fertilizer treatment significantly increased the content of soluble solids and volatile substances in fruits [67]. The mechanism was that microbial fertilizer increased the expression levels of AAT1, AAT2, LOX, ADH and HPL in fruits [68]. Water plays an important role in plant production, primary and secondary metabolism, so many authors have studied the effects of water limitation on aroma compounds, and these studies have focused on grapes [63]. The study found that reducing water and nitrogen supply before harvest can increase the VOCs content of grapes [69]. The pre-harvest bagging technology is to bag the fruit outside, and after a period of cultivation on the tree, the bag is removed or harvested together with the bag in the early stage of fruit harvest [5]. The content of total aroma substances and the production of ethyl butyrate and hexyl acetate decreased in bagged ‘Nanguo’ pear [70]. By summarizing the effects of cultivation and management conditions on pear and other fruit aroma substances, it was found that appropriate soil, fertilizer and water management systems will help produce more valuable fruits.’

Line395-401: ‘Pears produce different characteristic aroma compounds at different maturity stages. In the early stage of ‘Nanguo’ pear ripening, the content and species of aldehydes were the most, and the content and species of esters increased with fruit ripening. In addition to the fruit aroma, the early-harvested 'Nanguo' pears also mixed with some grass aroma due to its high hexanal content. The late-harvested 'Nanguo' pear had increased esters and showed strong fruit aroma [19].’

Line423-426: ‘There was a positive correlation between ethylene production and primary volatile esters, indicating that ethylene was a contributor to the induction of aroma-related esters in 'Nanguo' pear [74]. After low temperature treatment, ethylene production is low, causing loss of volatile esters [73].’

Comment 7: Line 479 - please use small letters 

Response: Thanks for your suggestions. We have corrected in the manuscript as your suggested.

Reviewer 4 Report

It is an interesting work for which I congratulate the authors

Recommendations:

Introduction - I suggest an improvement

The Extraction Identification Method of ‘Nanguo’ Pear -  For what? it is a little generalized

Category and Composition of Main Aroma Components in ‘Nanguo’ pear - I suggest an improvement considering the type of article

Figure 1. -  source

Amino Acid Pathway  - I suggest a more suggestive title

Carbohydrate Pathway - I suggest an improvement

Figure 2. -  source

Summary and Out Look - I suggest an improvement, a title that conveys more

I think it needs to be improved from a scientific point of view considering the type of article "Review"

Author Response

Review 4

It is an interesting work for which I congratulate the authors

Recommendations:

Comment 1: Introduction - I suggest an improvement

Response: Thanks for your comment. We have modified this part.

Line24-27: ‘Plants have the ability to synthesize, accumulate and release volatiles [1]. For plants, the release of volatile substances may be to spread seeds. For humans, these volatile compounds (VOCs) may have antibacterial or anticancer activity. People will also extract some of these ingredients to make perfume [2].’

Line40-46: ‘Considering the important influence of VOCs on flavor, a lot of research has been carried out on the VOCs of fruits. Breakthroughs have been made in the biosynthesis path of fruit aroma substances, the identification of key structural genes, and the regulation of exogenous hormones on the metabolism of aroma substances [1,11,12]. With the emergence of precision instruments such as gas chromatography-mass spectrometry (GC-MS), the research on aroma volatiles has become more in-depth, and the method for determining aroma volatiles has gradually matured.’

Line59-62: ‘Nanguo’ pear is a good example to study fruit aroma. So far, 79 volatile chemicals have been observed in ‘Nanguo’ pear. These compounds include esters, alcohols, ketones, terpenes, aldehydes and benzene compounds [8]. The types and concentrations of these volatiles vary according to their cultivation conditions and maturity [15,19].’

Comment 2: The Extraction Identification Method of ‘Nanguo’ Pear -For what? it is a little generalized

Response: Sorry for confusing you. We have modified this part.

Line83-84: ‘Each of them has its own characteristics and can be used selectively to obtain better extraction results.’

Line135-143: ‘Using a single method to extract aroma substances often cannot meet the needs of analysis. Currently, there is no good method to accurately reflect the actual aroma components or their proportions in fruits. The commonly used HS-SPME method cannot extract carboxylic acid compounds well, while the SDE can detect carboxylic acid compounds well, so SDE and HS-SPME can be used to analyze more complete aroma components [30]. When detecting the aroma components of ‘Nanguo’ pear, SDE can obtain more complete aroma information, but because of its solvent pollution, HS-SPME method is more used [23]. It is very important to find a faster and safer aroma determination method.’

Comment 3: Category and Composition of Main Aroma Components in ‘Nanguo’ pear - I suggest an improvement considering the type of article

Response: Thank for the suggestion of improvement. We have rewritten the relevant content.

Line191-193: ‘The content of aldehydes in early harvested ‘Nanguo’ pear increased, and its primary aroma components did not change, which may be caused by the short harvest interval [19]. ’

Line199-203: ‘We detected the VOCs of ‘Nanguo’ pear within 15 days after harvest (DAH). Among them, ethyl butyrate, ethyl hexanoate and hexyl acetate are considered to be the most important typical aroma substances based on their high content and low odor threshold, which are used to characterize the aroma content of ‘Nanguo’ pear (Table 1.) [28].’

Line208-210: ‘However, the synergistic effect between aromas is not clear, and it is not possible to determine the relationship between the strong aroma and the synergistic effect of aroma in ‘Nanguo’ pear.’

Line 213-214: ‘Currently, it is believed that the aroma of ‘Nanguo’ pear comes from the typical aroma substances.’

Comment 4: Figure 1. -source

Amino Acid Pathway - I suggest a more suggestive title

Carbohydrate Pathway - I suggest an improvement

Response: We refer to the article ‘El Hadi, M.; Zhang, F.-J.; Wu, F.-F.; Zhou, C.-H.; Tao, J. Advances in Fruit Aroma Volatile Research. Molecules 2013, 18, 8200–8229, doi:10.3390/molecules18078200.’ to draw Figure 1.

Thank you for your good suggestions, now we change the title to ‘Amino Acid Pathway of Ester Biosynthesis.’

We have rewritten ‘Carbohydrate Pathway’ and make it clear.

Line 306-308: ‘Another pathway is that pyruvate forms acetaldehyde under the action of PDC, then acetaldehyde is reduced to ethanol under the catalysis of ADH, and then ester is synthesized, but the pathway has not been confirmed.’

Line 319: The MVA pathway is carried out in the cytoplasm MEP pathway is in the plastid [58].

Comment 5: Figure 2. -source

Summary and Out Look - I suggest an improvement, a title that conveys more

I think it needs to be improved from a scientific point of view considering the type of article "Review"

Response: We refer to the article ‘El Hadi, M.; Zhang, F.-J.; Wu, F.-F.; Zhou, C.-H.; Tao, J. Advances in Fruit Aroma Volatile Research. Molecules 2013, 18, 8200–8229, doi:10.3390/molecules18078200.’ to draw Figure 2.

Thanks for your constructive suggestion. We have rewritten the relevant content.

Line 587-608: ‘The production of volatiles is affected by many factors. So far, our understanding of how these factors interact is limited. Currently, the contribution of volatiles to aroma is evaluated by OAV, but the matrix used in OAV calculation is mostly water, which is different from the real matrix of aroma, and the contribution to aroma cannot be well expressed [113]. There are many studies on bound aroma substances in grape and strawberry fruits, but few in apple, pear and other fruits [41,114]. The contribution of bound aroma substances to aroma is not clear. Although the characteristic volatile compounds in ‘Nanguo’ pear have been identified and quantified, the research on the biosynthetic pathway and precursor verification of their volatiles is still limited. For example, the biosynthetic pathway of carbohydrates in ester aroma volatiles is still unclear. At the late ripening stage of ‘Nanguo’ pear, the content of ethanol and acetaldehyde in the fruit increased rapidly [79]. Whether these ethanols will participate in the biosynthesis of esters as precursors is unknown. Most studies have focused on the biosynthesis of volatile aroma, and few researchers have studied the hydrolysis of esters. CXE is an esterase, but its substrate in ‘Nanguo’ pear is not clear. It was found that 1-MCP treatment significantly inhibited ADH activity, indicating that ADH may be regulated by ethylene, but its molecular mechanism is not clear, and more research is needed from the translation and post-translation levels [79]. When using plant hormones to treat ‘Nanguo’ pear, it was found that ethylene seemed to be an important participant in regulating the biosynthesis of fruit aroma, but its specific molecular mechanism was not clear enough. In the future, a plant hormone crosstalk network with ethylene as the core may be constructed.’

Reviewer 5 Report

The manuscript "Advances in aroma volatile research of ‘Nanguo’ pear" is dealing with typical fruit from Liaoning Province which possesses attractive aroma which changes during fruit ripening and storage. The manuscript is very interesting, well written and therefore I suggest acceptation in present form.

Author Response

Review 5

Comment 1: The manuscript "Advances in aroma volatile research of ‘Nanguo’ pear" is dealing with typical fruit from Liaoning Province which possesses attractive aroma which changes during fruit ripening and storage. The manuscript is very interesting, well written and therefore I suggest acceptation in present form.

Response: Thanks, we really appreciate your scrupulous review and kind corrections to our manuscript.